# Use of Brewers' Spent Grains as a Potential Functional Ingredient for the Production of Traditional Herzegovinian Product Ćupter

**Anita Lalić [1], Andrea Karlović [1] and Marina Marić [2,3,*]**

[1] Faculty of Agriculture and Food Technology, University of Mostar, Biskupa Cule b.b.,
80 000 Mostar, Bosnia and Herzegovina

[2] Herkon d.o.o. Mostar, Biskupa Cule b.b., 80 000 Mostar, Bosnia and Herzegovina

[3] Department of Chemistry, Faculty of Science and Education, University of Mostar, Matice Hrvatske b.b.,
88 000 Mostar, Bosnia and Herzegovina

* Correspondence: marina.maric155@gmail.com

**Abstract:** Ćupter is Herzegovinian candy made of must and flour/semolina. Much research about the incorporation of brewers' spent grains into the human diet has been published. The purpose of this study was to partially replace semolina (Samples 1 and 2) and flour (Samples 3 and 4) with brewers' spent grains originating from industrial (Samples 1 and 4) and craft breweries (Samples 2 and 3) and study nutritive, chemical, and preference properties of the product. In this research, the authors aimed to find application of this already proven functional ingredient in ćupter production. Values for pH were higher for all samples compared to the traditional recipe. Samples produced with flour had higher values of water activity (0.86 ± 0.01) and moisture (41.82 ± 1.68 and 41.11 ± 1.41). Ash content increased with BSG addition, but between samples, there were no significant differences. Collected data showed significant differences in fat levels. Higher protein content was measured for Samples 4 (6.60 ± 0.17) and 1 (6.13 ± 0.07). The highest total sugar content was measured for Sample 1. The general appearance for all samples was "moderately like". Nutritive value was improved with the addition of BSG, but recipes and drying should be modified to improve consumer acceptance.

**Keywords:** brewers' spent grain; functional food; ćupter; traditional product; grape must

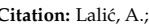



## 1. Introduction

Functional food, in the last decade, has represented a wide field of research [1], primarily due to the positive impact on human health [2]. Namely, functional food is a food that not only aims to nourish the population, but its composition has a beneficial effect on human health, i.e., prevents the development of certain diseases and/or has positive effects in eliminating them [3,4]. One of the five approaches claims that functional food can be made if the concentration of the component, which is naturally present in the food, is increased [4,5]. More precisely, if traditional food is enriched with present components, either by changing the recipe or the production process, it can be declared as a functional food. In the traditional approach to functional ingredient discovery and characterization, health claims are assigned after bioactivity has been established. The authors of this paper are referring to a new approach in functional ingredient discovery, i.e., artificial intelligence that does not start with a health benefit but that rather seeks application after identifying and characterizing bioactive substances [6]. The traditional Herzegovinian product ćupter can be described as sweet jelly and is made of grape must and semolina or flour, with the possible addition of some nuts depending on preferences. Ćupter is made in the area of Brotnjo, a well-known wine-growing region. To this day, the preparation of ćupter is mainly homemade by grandparents, which is opening space for greater production and recognition as a local brand. The sub-Mediterranean climate enables the cultivation of the

traditional varieties Zilavka and Blatina, from which top-quality white and red wine is produced. Ćupter, after achieving the desired consistency and drying, can be consumed for a longer period.

This article aims to present the use of brewers' spent grains (BSG) as a potential functional food ingredient in the traditional Herzegovinian product ćupter. Specifically, one part of semolina and flour were replaced by a by-product of the brewing industry, the BSG. The traditional ćupter, with the use of BSG, becomes a functional food based on the content of the proteins. Traditionally, ćupter has a low concentration of protein, but with the addition of BSG as a rich source of protein, it is expected that the concentration will grow, and it can be considered a functional food.

There are a few reasons why BSG was chosen as the replacement of flour and semolina. Primarily, the desire was to find a way to reuse BSG, which is lagging behind in Bosnia and Herzegovina as a by-product of the brewing industry. Unfortunately, in Bosnia and Herzegovina, BSG is mostly dumped in fields and burned, or at best sold as animal feed [7]. Given the good nutritional composition and low purchase price, it can be used in the food industry. However, Sahin et al. [8] showed that BSG, due to its high fiber and protein source, can be a good replacement for flour and semolina in pasta, which led to the idea to try it in the traditional Herzegovinian product ćupter as a replacement of flour and semolina.

## 1.1. Brewers' Spent Grain

BSG represents insoluble components left after lautering in the brewing industry. BSG is the most important by-product of the brewery [9–11], the annual production of which is estimated to be ~39 million tons, with ~3.4 million tons produced in the European Union [12].

BSG is a lignocellulosic material that is rich in protein (20%), dietary fiber (70%), minerals, essential fatty acids, and vitamins [11–17], the chemical composition of which can vary depending on several factors, such as the quality of barley or other cereals used in beer production, harvesting time, malting germination conditions, and the quality of unsweetened raw materials [10,11]. Since BSG separates after mashing [10], it is expected that the moisture is between 75 and 80% [11,14,17]. Due to the water content, one of the major problems is the process of drying and storage. Dried spent grains should contain no more than 10% water, which prevents spoilage but also reduces the volume of the BSG [7,11].

Since the BSG is relatively affordable, low cost, and nutritionally rich, it can be used for the preparation of possible functional food. Previous studies have been reported concerning the identification of specific BSG-derived bioactive peptides (Cermeno et al., 2019) and referring to health claims [18]. In terms of this, brewers' spent grains are a great source of protein but are still not yet well involved in a diet [13]. The use of food processing by-products (FPBs), as Kim et al. [19] state, can save the high cost of post-treatment processes, and bring not only a cheap source for the new product but also prevent environmental disasters. Based on this and on the fact that Bosnia and Herzegovina has a growing beer industry and so BSG lags behind like waste, BSG was used in this research. Nevertheless, considering the desirable nutritive and functional properties, BSG use in human nutrition is recommended to provide health benefits [15,20–22].

Furthermore, the paper compares craft and industrial BSG. Namely, it is known that although there is no significant difference in chemical and nutritional composition between craft and industrial BSG, the degree of grinding is different because craft brewers use different types of milling equipment due to lower capital for investment in equipment [23]. More precisely, craft BSG is larger, sometimes with whole grains, while industrial BSG is finely ground and resembles bran. Therefore, this paper sought to see whether different degrees of grinding affect consumer preferences.

*1.2. Must*

The process of making wine begins with crushing and pressing the grapes to obtain the grape juice—the so-called must. The process of pressing separates berries from the petioles. This process is followed by grinding by which the grapes are crushed and the juice or must is extracted by pressing. Before the beginning of alcoholic fermentation, freshly squeezed grape juice is used to prepare ćupter. Based on Nurkovic [24], the concentration of alcohol in the must is not higher than 1%.

The Herzegovina region has one indigenous sort of white wine named Zilavka (white type), and it is the most abundant type of wine with a share of 70%. The influence of the sub-Mediterranean climate with lots of sunny days and calcareous soils provides favorable conditions for grapevine cultivation [24,25].

Grapes and grape must have a high concentration of carbohydrates, more precisely sugars (approx. 17 g/100 g). Because of sugars, there is a high caloric content (65 kcal/100 g) and a relatively low glycemic index that causes grapes and grape products to be beneficial for a healthy diet. Grape must is rich in minerals such as manganese, potassium, sodium, and iron and is also rich in vitamins such as A, B1, B2, B6, and C [26,27].

## 2. Methods

*2.1. Materials*

The control sample traditionally prepared (TC) was homemade and donated for the purpose of this research. The traditional recipes for ćupter made from white grape must + flour and white grape must + semolina were modified; i.e., 62% of semolina and 37% of flour were replaced with BSG. Semolina (Franck, Zagreb, Croatia) and white flour (Klas, Sarajevo, Bosnia and Herzegovina) used are commercially available and were bought in a local supermarket in Bosnia and Herzegovina. In the preparation of ćupter, must was used mostly from the white grape variety of Žilavka mixed with Smederevka, Krkošija, and Dobrogoština sorts of must. It was obtained from the AgroOdak d.o.o., a local producer of wine from Čitluk, Bosnia and Herzegovina. Craft BSG (CBSG) was obtained from Trojanska pivovara, a local brewery located in Čapljina, Bosnia and Herzegovina. Industrial BSG (IBSG) was obtained from Hercegovačka pivovara, an industrial brewery from Mostar, Bosnia and Herzegovina. Both were generated from a process using 100% malted barley.

Procedure for Preparation of Traditional and Ćupter with Added Industrial and Craft BSG

Modified ćupters with IBSG, CBSG, semolina, and flour were prepared. The white grape must was heated and filtered to remove yeast and other mechanical impurities. After that, IBSG or C BSG was added with flour or semolina according to the masses specified in Table 1. The mixture for ćupter was mixed with a whisk all the time to prevent the forming of lumps. After they thickened, samples were then poured into shallow containers (approximately 20 cm in diameter and 2 cm in height) and allowed to cool to room temperature for 8 h. They were then removed from the dishes and placed to dry in a solar drier (150 cm wide, 50 cm deep, and 70 cm high), which was equipped with two fans that enabled proper drying of the ćupter. All four samples (1, 2, 3, and 4) were prepared in triplicate in September 2020 at the Faculty of Agriculture and Food Technology, University of Mostar (Figure 1), and after drying stored in a dry and cold place wrapped in vacuum bags.

**Table 1.** The recipe used for preparation of traditional and modified traditional ćupter.

| Sample | Recipe for Ćupters |
|---|---|
| 1 | 180 mL of white grape must + 18 g of IBSG + 11.2 g of semolina |
| 2 | 180 mL of white grape must + 18 g of CBSG + 11.2 g of semolina |
| 3 | 180 mL of white grape must + 13.5 g of CBSG + 22.5 g of flour |
| 4 | 180 mL of white grape must + 13.5 g of IBSG + 22.5 g of flour |

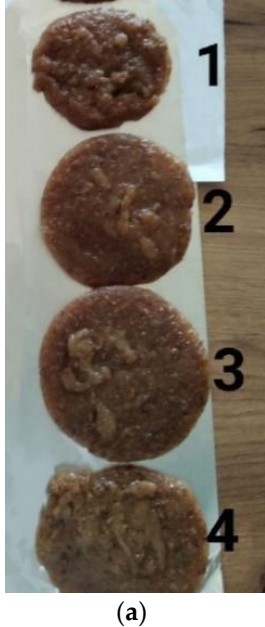 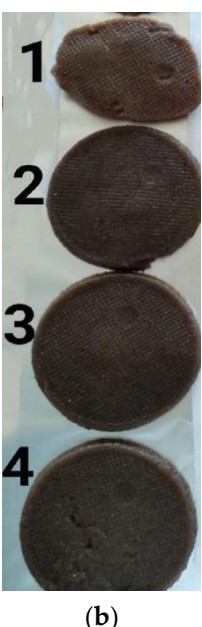

(**a**)             (**b**)

**Figure 1.** Visual appearance of (**a**) Samples 1, 2, 3, and 4 before solar drying, and (**b**) Samples 1, 2, 3, and 4 after solar drying.

Table 1 shows 4 samples that have two different recipes. Both recipes, with the addition of flour or semolina, were developed based on a traditional recipe and the different thickening abilities that flour and semolina have. Due to the different proportions of flour or semolina, different proportions of IBSG and CBSG were obtained. About 37% IBSG or CBSG was added to the flour recipe, while about 62% IBSG or CBSG was added to the semolina recipe.

### 2.2. Nutritive and Chemical Analysis of Ćupter

Samples of ćupter (500 g per sample) were homogenized using a Grindomix GM200 knife mill (Retch, Haan, Germany) to obtain homogeneous samples. Nutritive and chemical analyses were performed by applying ISO methods and standard analytical methods. All measurements were done in 3 replicates using chemicals of analytical grade.

### 2.2.1. pH Determination

The method of Nicolai et al. [13] was used for the determination of pH. The sample was ground in a Grindomix GM200 knife mill (Retch, Haan, Germany). A reference sample of 10 g was taken from each of the four samples of ćupter and diluted with 100 mL of distilled water. After mixing on a chemical shaker for 10 min and stabilization of suspension, pH measurements were done in triplicate using a Hach (HQ11D Digital) pH meter.

### 2.2.2. Water Activity (Aw) Measurement

HygroPalm (Rotronic, Switzerland) was used to determine the activity of water in all prepared samples of ćupter. This value is very important for food safety and especially in terms of bacteria and yeast growth, and mold-free shelf life (MFSL) [28]. The samples were ground and placed in special cups of the device up to the indicated edge. The activity was measured daily during the drying of the samples. All these measurements were done 3 times.

### 2.2.3. Moisture Determination

Determination of water was performed by drying the final product ćupter in the oven (Binder FD 23, GmbH Memmert UF75 Plus, Schwabach, Germany) at 103 ± 2 °C according to BAS ISO 6496:2008. Each of the ćupter samples was weighed to the nearest 1 mg in

quantities of 5 g. They were dried in vegeglas bowls together with glass rods. The samples were weighed to a constant mass. Repetitions were done in triplicate, and the end result was expressed in percentages.

### 2.2.4. Ash Determination

Ash content was obtained by use of the gravimetric method according to BAS ISO 5984:2008, implementing a muffle furnace at 550 °C. All measurements were made in triplicate and expressed as percentage.

### 2.2.5. Protein Determination

The crude protein content was determined by the Kjeldahl method (Kjeltec™ 8200, Foss), according to BAS EN ISO 5983-1/Cor1:2012 and BAS EN ISO 5983-2:2010, which involves the destruction of organic matter at 420 °C with the use of a block digestion unit (Foss—DT 208 and 220 Digestor™). This method involves determining the nitrogen content, which is then multiplied by a conversion factor of 6.25 to obtain the protein content.

### 2.2.6. Fat Determination

The crude fat content was assessed according to the BAS ISO 6492:2008 method of determining the fat content that includes fat hydrolysis and fat extraction with petroleum ether on the extraction device (Soxtherm 2000, Gerhardt, Germany).

### 2.2.7. Total Sugar Measurement

The total sugar content was determined using the Luff Schoorl method. Firstly, for the determination of reducing sugars, a stock solution was prepared. The basic filtrate of samples was then prepared from the stock solution by the addition of Carezz I prepared with zinc acetate dihydrate (Fisher Scientific, Loughborough, UK), glacial acetic acid (Sigma Aldrich, Saint Louis, MO, USA), and Carezz II prepared with potassium hexacyanoferrate(II) trihydrate (Centrohem, Stara Pazova, Serbia) solution. A basic filtrate made for the determination of reducing sugars was further used for the determination of total sugars. Luff Schoorl solution prepared with citric acid (Merck, Darmstadt, Germany), copper sulfate pentahydrate (Gram mol, Zagreb, Croatia), and sodium carbonate (Kemika, Zagreb, Croatia) was used for the reflux heating part of the prepared samples. For this titrimetric method, sodium thiosulfate (Kefo, Sarajevo, B&H) was used together with starch (Lachner, Neratovice, Czech Republic) as an indicator. In this step, acid hydrolysis of disaccharides and polysaccharides to produce monosaccharides was done using hydrochloric acid 37% (Carlo Erba, Emmendingen, Germany).

### 2.2.8. Energy Kcal

An IKA WERKE C 5000 calorimeter was used for caloric determination of prepared ćupter samples. About 0.5 g of the homogenized sample was placed in a crucible and then a decomposition vessel. The sample placed this way is in a so-called "calorimetric bomb". At the end of the experiment, the results were displayed on the instrument as cal/g. The obtained results were translated into kcal and by multiplying by a factor of 4.1868 in kJ as well.

### 2.2.9. Chemical Analysis of White Grape Must

The chemical analysis of white grape must (pH, ethanol, reducing sugars, and total acids) was conducted by Foss (OenoFoss™, Hilleroed, Denmark). On the principle of FTIR technology, with a small amount of sample (approx. 1 mL) and a short analysis time, the results for the above parameters were obtained. The protein content was conducted as previously described in Section 2.2.5 Protein Determination.

2.2.10. Preference Test Determination

For the preference test, there were 60 panelists from the University, mostly students. The procedure for the preference test determination was common for testing. The room was prepared in advance with samples. Every panelist obtained their own place with a sample, after which the procedure was explained. Panelists answered the preference sensory questions via Google Forms during the testing of individual ćupter. The information about panelists is shown in Table 2.

**Table 2.** Panelist information.

| Gender | Age | Trained (Yes/No) | Student/Faculty Employee |
|---|---|---|---|
| M → 8.5% <br> F → 91.5% | <20 → 0% <br> 20–30 → 100% <br> >30 → 0% | yes → 100% <br> no → 0% | Student → 93.5% <br> Faculty employee → 6.5% |

*2.3. Data Analysis*

All measurements for nutritive analysis of ćupter were done in triplicate and shown as mean ± SD. Statistical determination was done using SPSS Statistica with one-way ANOVA and post hoc multiple comparisons Tukey tests (significance level: $p < 0.05$) to see where the differences between samples occurred. For the preference test descriptive statistic, frequencies were used. Spearman correlation tests were also used to show the possible connections in preference evaluation.

**3. Results and Discussion**

*3.1. Nutritive Value*

The results of analysis TC with ćupter with BSG (Samples 1, 2, 3, 4), and raw materials (BSG and grape must) are shown in Tables 3–5, respectively.

**Table 3.** Results of nutritive and chemical composition of TC and ćupter with BSG (Samples 1, 2, 3, 4), mean ± standard deviation values.

| Sample | pH | Water Activity (%) | Moisture (%) | Ash (%) | Protein (%) | Fat (%) | Total Sugars (%) | Energy kcal/kJ |
|---|---|---|---|---|---|---|---|---|
| TC | 3.56 ± 0.01 | 0.89 ± 0.01 | 27.08 ± 0.54 | 0.68 ± 0.01 | 0.67 ± 0.01 | 0.10 ± 0.03 | 52.25 ± 0.32 | 212.9/891.4 ± 0.87 |
| 1 | 4.11 ± 0.01 | 0.73 ± 0.01 | 24.02 ± 1.52 | 1.50 ± 0.39 | 6.13 ± 0.07 | 0.75 ± 0.03 | 51.07 ± 0.83 | 275.3/1152.6 ± 0.22 |
| 2 | 3.94 ± 0.005 | 0.71 ± 0.01 | 21.92 ± 0.69 | 1.87 ± 0.37 | 5.95 ± 0.02 | 0.44 ± 0.04 | 37.43 ± 0.87 | 277.1/1160.2 ± 0.27 |
| 3 | 3.90 ± 0.01 | 0.86 ± 0.01 | 41.82 ± 1.68 | 1.41 ± 0.10 | 5.93 ± 0.12 | 1.31 ± 0.01 | 39.66 ± 0.72 | 200.1/837.8 ± 0.65 |
| 4 | 3.86 ± 0.01 | 0.86 ± 0.01 | 41.11 ± 1.41 | 1.42 ± 0.10 | 6.60 ± 0.17 | 0.47 ± 0.01 | 40.37 ± 0.77 | 200.0/837.4 ± 1.15 |

**Table 4.** Chemical analysis of white grape must, mean ± standard deviation values.

| Sample | pH | Reducing Sugars (Glucose and Fructose) | Total Acids | Ethanol vol% | Protein (%) |
|---|---|---|---|---|---|
| White grape must | 3.58 ± 0.01 | 240 ± 0.01 | 3.7 ± 0.06 | 0.2 ± 0.5 | 0.66 ± 0.007 |

**Table 5.** Chemical analysis of IBSG and CBSG (calorific value of dried CBSG and IBSG), mean ± standard deviation value.

| Sample | Moisture (%) | Ash (%) | Fat (%) | Proteins (%) | Total Sugars (%) | Energy kcal/kJ |
|---|---|---|---|---|---|---|
| CBSG | 66.30 ± 0.75 | 0.90 ± 0.05 | 1.67 ± 0.14 | 13.03 ± 0.19 | 10.33 ± 0.32 | 186.9/782.6 ± 0.30 |
| IBSG | 72.75 ± 0.30 | 1.06 ± 0.03 | 2.97 ± 0.04 | 12.76 ± 0.19 | 5.04 ± 0.42 | 184.6/772.6 ± 0.25 |

In this work the major by-product of the brewing industry, BSG, was used to enrich traditional recipes with an increased potential nutritive value, more precisely increased protein content. The aim of this article was to determine the chemical and nutritive composition of traditionally prepared ćupter and ćupter with added BSG. In the control sample—traditional ćupter—as in all samples with IBSG and CBSG, the pH, water activity, moisture, ash, proteins, fats, total sugars, and caloric value were determined. In addition, analyses of IBSG and CBSG and grape must were made.

Since BSG is added in relation to the traditional preparation of ćupter, different nutritive and chemical values are expected. Recipes for traditional ćupter differ depending on whether semolina or flour are used as ingredients, since their moisture absorption capacity is different. The aim of the research primarily went in two directions: to compare the share of protein in traditional ćupter and ćupter with the addition of BSG with an emphasis on obtaining a new functional product, and to see if there are chemical and nutritive differences between ćupters with the addition of IBSG and CBSG.

### 3.1.1. pH

The following ranges were observed: pH value ranged between 3.56 and 4.11 and showed a statistically significant difference between every sample. Based on Racchi et al. [29], the pH value and the activity of water in food are the basic parameters that determine the perishability of food, that is, the possibility of the appearance of mold, bacteria, and yeast. Optimizing the pH value and activity of water affects the storage of a product and predicts its shelf life.

### 3.1.2. Water Activity and Moisture

Samples 1 and 2, due to the values of water activity of $0.73 \pm 0.01$ and $0.71 \pm 0.01$, could allow the growth of xerophilic fungi (*Aspergillus chevalieri* and *Aspergillus candidus*) according to the literature. Samples 3 and 4 had slightly higher values of water activity of $0.86 \pm 0.01$ each.

According to the literature, these values allow the growth of bacteria (*Staphylococcus aureus*) and of yeasts (*Debaryomyces* spp.). These values of water activity will not inhibit the growth of the aforementioned fungi, bacteria, and yeast, but the combination of adequate pH values and water activity may be the subject of further research and optimization of the final product to avoid that impact [29].

Semolina is made of *Triticum durum*, while flour (farina) is made of *Triticum aestivum*. Due to the difference in the variety of cereals, semolina and flour have different abilities to thicken and absorb water [30]. Based on this, less semolina was added to the basic recipe (Samples 1 and 2) compared to flour (Samples 3 and 4). Namely, semolina absorbs water faster and swells, which is why a smaller amount is needed. In view of the above, Samples 3 and 4 had a higher mass and volume (thicker than 1 and 2), which is why they kept a higher moisture content during the same drying time at the same temperature. However, in order to ensure a microbiologically correct product for a longer period of time, it is necessary to reduce its water activity. The above values for activity show that they allow the growth of fungi, bacteria, and yeasts. One way to achieve lower water activity is by additional drying of samples for recipes of Samples 3 and 4.

### 3.1.3. Ash Determination

The ash content varied between 0.68 and 1.867% in control and other samples. There was a statistically significant difference in the control sample and all other samples, but no statistically significant difference between Samples 1, 2, 3, and 4, which was expected given that the control sample had no BSG additive.

### 3.1.4. Protein

The addition of BSG increased the protein content from 0.67% in traditionally prepared (control sample) ćupter to 6.13% in Sample 1, 5.95% in Sample 2, 5.93% in Sample 3, and

6.60% in Sample 4. A statistically significant difference in protein values was found with respect to the recipe ($p < 0.05$). IBSG samples (Samples 1 and 4) had higher protein content. The mentioned results confirmed the initial assumptions of this research that the addition of BSG will significantly increase the protein content in the new product, and the use of IBSG and CBSG will also contribute to the difference in protein content.

The first thesis, that the protein content will increase, can be explained by the fact that the BSG used contains 13.03 (industrial) and 12.76 (craft) protein content, respectively. Since the change of traditional recipes for ćupter resulted in enrichment with proteins and hence amino acids, it can be declared as a functional food [4].

The second thesis, that the use of IBSG or C BSG can contribute to the difference in the protein content, was confirmed by the results obtained, which showed consistency. Namely, samples with the use of IBSG (1 and 4) showed a higher share of total protein, which was shown also by analysis of the raw material BSG. There could be a few reasons for this. One is the quality of malt used. Poor quality malt with insufficient modification contains a higher level of residual protein [31]. In general, in beer production, high protein content is undesirable, but in this case, it is the opposite. This higher content could be also explained due to the different mashing processes in industrial vs. craft breweries, since proteins of the grains are also altered depending on this [32]. In addition to the high protein content, the high content of essential amino acids (approx. 30%) in BSG should also be emphasized with particular interest to lysine, as it is generally the limiting amino acid in cereal-based foods [33].

### 3.1.5. Energy

A statistically significant difference in energy values was found with respect to the recipe ($p < 0.05$). The Tukey test revealed statistically significant differences in energy values between traditionally prepared ćupter and every sample. The addition of 18 g of I BSG (Sample 1) and 18 g of C BSG (Sample 2) increased the total energy value by 29.3% for Sample 1 and 31.15% for Sample 2 compared to traditionally prepared ćupter. In addition, increasing the BSG from 13.5 g to 18 g in Sample 1 and Sample 2 increased the energy value by 37.6% for Sample 1 and 38.5% for Sample 2. The rest of the energy value in kcal consists of dietary fiber, which is especially present in BSG [34].

The shortcoming of this work can be seen in the share of fiber, i.e., future research should include the determination of total fiber, which would give a complete picture of the nutritive value of the product.

### 3.1.6. Fat

Ćupter is considered as a low-fat food; measured fat content (%) for TC was $0.10 \pm 0.03$. Sample 3 had the highest value of fat ($1.31 \pm 0.01$) and showed a statistically significant difference in fat compared to TC, Samples 1, 2, and 4. This sample obtained the highest flavor grade in comparison to other samples. Fat plays a role in determining the desirable sensory attributes of food products and affects the acceptability and enjoyment of food [35].

### 3.1.7. Total Sugars

Sample 1 has the highest total sugar content ($p < 0.05$) compared to Samples 2, 3, and 4. The Luff Schoorl method was not a sensitive enough method, and more reliable methods should be used in future research. Based on the obtained results, we can conclude that the obtained ćupters are a very good source of total sugars and high energy.

### *3.2. Preference Test*

Ćupter is a very pleasant sweetish taste that is consumed usually as a dessert together with almonds, walnuts, and dried figs. It has no additives, chemicals, or refined sugar in it [36]. The addition of BSG can influence the odor, color, texture, and flavor profile of the traditional product, the taste of which is well known to consumers. Therefore, preference

testing is necessary to validate consumer acceptance of newly developed ćupter with the addition of BSG [37,38].

The 60 panelists evaluated the four different receipts for ćupter using an 8-point Likert questionnaire scale that covered the preference analysis of color, texture, smell, taste, and general appearance. This scale included 1—like extremely, 2—like very much, 3—moderately like, 4—slightly like, 5—neither like nor a dislike, 6—moderately don't like, 7—dislike very much, and 8—dislike extremely. In this way, we could see which ćupter was most acceptable in terms of preference, as well as the direction to improve the final product. The obtained values were translated into numerical form and are shown as mean ± SD in Table 6. The preference test will enable the improvement of the final product.

**Table 6.** Preference test of ćupter mean ± standard deviation value.

| Sample | Color | Texture | Scent | Flavor | General Appearance |
|--------|-------|---------|-------|--------|--------------------|
| 1 | 4.17 ± 1.69 | 3.05 ± 1.79 | 3.43 ± 1.73 | 3.45 ± 1.89 | 3.20 ± 1.61 |
| 2 | 2.90 ± 1.53 | 3.25 ± 1.76 | 3.60 ± 1.66 | 3.56 ± 1.82 | 3.13 ± 1.84 |
| 3 | 3.30 ± 1.77 | 3.15 ± 1.76 | 3.40 ± 1.82 | 3.83 ± 1.65 | 3.45 ± 1.75 |
| 4 | 3.47 ± 1.70 | 2.71 ± 1.60 | 3.45 ± 1.68 | 3.37 ± 1.91 | 3.20 ± 1.46 |

The best grade for color was given to Sample 1 prepared with IBSG, which had the darkest color. This is related to the IBSG, which has a different degree of milling relative to CBSG (Sample 2 and 3). In addition, Hejna et al. [39] showed that the particle size of BSG strongly influences the darker color. Compared to Sample 4, which was also made by the addition of IBSG, Sample 1 had a darker color because it had a different ratio of IBSG (Table 1).

In addition, the Spearman correlation test showed the mutual strength of the connection between color, texture, smell, and taste with the general appearance. All of the aforementioned had a positive connection with general appearance. With a correlation of 0.584 for color, 0.733 for texture, 0.689 for scent, and 0.917 for flavor, the Spearman correlation test showed that there was a strong positive correlation with general appearance. The most affected on general appearance was flavor. In general, the texture of these samples received the lowest grade, especially Sample 4. As the texture is directly connected to the fiber content [8] and the preparation process, in the next research, special attention should be paid to the management of the fiber content and the preparation process itself.

The continuation of research in order to achieve a better product should go in the direction of reducing the amount of water to obtain a firmer structure and reduce the feeling of stickiness when consuming the product. The average general appearance for all samples was 3, which shows that panelists moderately liked the prepared samples.

Preference Test Results

The obtained values for the preference test were translated into numerical values and are shown as mean ± SD in Table 5.

## 4. Conclusions

The reuse of agricultural by-products is of great interest. The present study demonstrated that the use of BSG as a potential functional ingredient in ćupter production could be one of the ways to reuse this cheap and easily available by-product from the beer industry. Due to its high nutritive value with a special accent on high-quality protein content, its addition significantly improves the nutritive value of ćupter. Consumers want to benefit their health with the consumption of functional food, but another challenge left for future research is to improve general acceptance from "moderately-like" to "like extremely" or "like very much" by changing recipes. A few things can be reconsidered for improvement; due to easier and shorter production, the best option is to make only a combination with semolina and BSG (flour requires longer preparation time due to weaker and slower thickening, and there is a greater possibility of forming lumps that are undesirable in taste); due

to better shape and appearance, and easier handling during the drying process (samples in the research were slightly thinner than traditional), the proposal is to use approx. 300 mL of grape must for the above-mentioned shape dishes instead of 180 mL; due to better taste and preferences of the panelist and consumers, it would be good to add different nuts to the recipe.

**Author Contributions:** Conceptualization, A.L. and A.K.; formal analysis, A.L., A.K. and M.M.; software, M.M.; writing—original draft preparation, M.M. and A.K.; writing—review and editing, A.L.; supervision, A.L.; funding acquisition, A.L. and M.M. All authors have read and agreed to the published version of the manuscript.

**Funding:** This research was founded, as a part of the MOstarT Science 2021 Project, by the Western Balkans Alumni Association (WBAA), organisation funded by the European Union, Project number: WBAA2021013.

**Institutional Review Board Statement:** Not applicable.

**Informed Consent Statement:** Not applicable.

**Data Availability Statement:** For all additional data and materials, the corresponding author needs to be contacted.

**Acknowledgments:** The authors are grateful for the support of Herkon d.o.o. Mostar during the experimental analysis for this work. The authors wish to thank Trojanska and Hercegovačka brewery for the donation of BSG and Agroodak for the donation of grape must.

**Conflicts of Interest:** The authors declare no conflict of interest.

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
