# Peer review of "Use of Brewers’ Spent Grains as a Potential Functional Ingredient for the Production of Traditional Herzegovinian Product Ćupter"

_fermentation, doi:10.3390/fermentation9020123_

Round 1
Reviewer 1 Report
The authors reported the use of brewer's spent grain in traditional Herzegovina candy Ćupter as one of the effective uses of brewing by-products. Effective utilization of brewing by-products is being researched all over the world, and it is an important research theme from the viewpoint of SDGs. However, this manuscript is less scientifically important experimental results.
Major point
- It is understandable that the addition of BSG increases the nutritional value, but there is no experimental result on the "functional ingredient" in the title. What are the ingredients in BSG that the author expected? And what are the expected benefits of eating this improved candy?  
- Sensory tests were performed on samples only. Comparison with TC is required.
Minor point
- The numerical value always precedes the unit, and a space is always used to separate the unit from the number (See, The International System of Units (SI)-8th edition-2006). % is also no exception.
- Do not abbreviate the first appeared word. 2P L1 BSG, 3P L7-8 CBSG, IBSG
Author Response
Dear reviewer,
Please see the attachment.
Marina Marić

Reviewer 2 Report
The manuscript "Use of Brewer’s Spent Grains as a Functional Ingredient for the Production of Traditional Herzegovinian Product Ćupter"
was submitted to Fermentation and focuses on making a functional food from a traditional local sweet.
The manuscript is in general easy to understand, and the results are clearly presented and discussed.Minor comments: Rephrase this sentence:
Ćupter can be described as sweet jelly, which is known only in one part of
Turkey (so-called pestil).
Rephrase this, kneading is a strange choice of words:
1.2. Must
The process of making wine begins by kneading and grinding the grapes to obtain the grape juice so-called must.
"Zilavka is an abundant type of wine with a share of 70% [21,22]". Where, in Herzegovina?
This sentence is too strongly worded. The research is still on-going whether wine is healthy at all. "Grape must and wine
are known as great sources of antioxidants that have a great effect on human health. It can protect from cardiovascular diseases and cancer [25]."
Chapter 2.1.1: approx., not app.
2.2.5. Protein Determination. "A factor of 6.25 was used". What factor, please write details.
This part is too casually written: "Foss needs a few drops of grape must
or wine after which we get the results on the PC screen in 3 min"
page 7:
"allows the growth of Xerophilic molds and Saccharomyces bisporus." Xerophilic molds should not be in italics! Saccharomyces bisporus does not exist, it is not an accepted name!!!
"According to the literature [27] these values allow the growth of molds: Staphylococcus aureus, Saccharomyces(bailli) spp., and Debatymoyces. "
This sentence is completely wrong, these are bacteria and yeasts, and Zygosaccharomyces bailii, and Debaryomyces. And Triticum aestivum, not aestium.
"Accordingly, it is concluded that samples 3 and 4 were dried for a short time."
What does this sentence mean?
Tuckay test????
Data Availability Statement: is missing!
Sensosry testing is not listed in Materials and Methods!!! This should be corrected.
Author Response
Dear reviwer,
Please see the attachment.
Kind regards,
Marina Marić

Reviewer 3 Report
This manuscript addresses an activated method of brewer's spent grain and may provide brewing industry with some hints of these methods. The authors prepare four prototypes, measured basic nutrition and carried out a preference test. However, some modifications will be needed for publication. There are few discussions in this manuscript, so I recommend that the authors should discuss the results.
- The authors mention the possibility of BSG in introduction with reference 17. However, I fail to understand why the authors select BSG and replace semolina and flour into BSG. Therefore, please explain the reasons in more detail. Also, please show the reasons why authors compare craft and industrial BSG.
- The authors prepare 2 types of ćupter in Table 1, semolina and flour. There are differences of formulation between semolina and flour. Please show the background of difference of the receipt.
- The authors perform sensory evaluation for 4 prototypes. I wonder if the evaluation could be a preference test for general consumer. I recommend that sensory evaluation should change preference test. And please show procedure of test and information of panelist including recruiting, gender, age and trained/non-trained in material and method section.
- The authors observe the differences of preference test, especially color in sample 1 and texture in 4. Could the authors discuss the reasons based on result of general components such as protein, sugar, Maillard reaction and so on?
- The authors point out modification of preference in future study. Please demonstrate result of traditional ćupter in the sensory evaluation and direction of the modification based on discussion about these qualities.
Author Response
Dear reviewer,
Please see the attachment.
Kind regards,
Marina Marić

Round 2
Reviewer 1 Report
The authors have added Ref. 31 to explain the functionality of traditional candies ćupter kneaded with brewer’s spent grains. A protein derived from brewer’s spent grains is expected to prevent diabetes by reducing insulin resistance. The importance of research on the effective use of brewer’s spent grains was mentioned in the previous review comment. I also understand that the manuscript fits the purpose of this special issue “Organic Waste Valorization into Added-Value Products”. but the functionality of this research is still questionable. The description of functionality depends on solely on previous reports and is not the result of human clinical trials. Therefore, it is strongly recommended to focus on the effective use of brewer’s spent grains by removing “as a Functional Ingredient” from the title.
Author Response
Dear Reviewer,
Please see the attachment.
Kind regards,
Marina Marić

Round 3
Reviewer 1 Report
The authors responded appropriately to the reviewer's comments. It recommend that this manuscript be published in Fermentation.
Author Response
Dear Reviewer,
Please see the attachment.
Kind regards,
Marina Marić
Corresponding author.
